# An In Vitro System for Evaluating Molecular Targeted Drugs Using Lung Patient-Derived Tumor Organoids

**DOI:** 10.3390/cells8050481

**Published:** 2019-05-20

**Authors:** Nobuhiko Takahashi, Hirotaka Hoshi, Arisa Higa, Gen Hiyama, Hirosumi Tamura, Mayu Ogawa, Kosuke Takagi, Kazuhito Goda, Naoyuki Okabe, Satoshi Muto, Hiroyuki Suzuki, Kenju Shimomura, Shinya Watanabe, Motoki Takagi

**Affiliations:** 1Medical-Industrial Translational Research Center, Fukushima Medical University, Fukushima 960-1295, Japan; tkhsnbhk@fmu.ac.jp (N.T.); h-hoshi@fmu.ac.jp (H.H.); arisaxon1945@gmail.com (A.H.); hiyamag@fmu.ac.jp (G.H.); tamura@fmu.ac.jp (H.T.); swata@mvc.biglobe.ne.jp (S.W.); 2Department of Bioregulation and Pharmacological Medicine, Fukushima Medical University, Fukushima 960-1295, Japan; shimomur@fmu.ac.jp; 3Research and Development, Biological Evaluation Technology 2, Olympus Corporation, Hachioji, Tokyo 192-8512, Japan; mayu_ogawa@ot.olympus.co.jp (M.O.); k_goda@ot.olympus.co.jp (K.G.); 4Research and Development, SSD Technology Innovation 3, Olympus Corporation, Hachioji, Tokyo 192-8512, Japan; ko_takagi@ot.olympus.co.jp; 5Department of Chest Surgery, Fukushima Medical University School of Medicine, Fukushima 960-1295, Japan; okaben@fmu.ac.jp (N.O.); smutoo@fmu.ac.jp (S.M.); hiro@fmu.ac.jp (H.S.)

**Keywords:** molecular targeted therapy, cancer immunotherapy, cancer immunity, molecular targeted drugs, antibody drug, antibody-drug conjugate, immune checkpoint inhibitor, patient-derived tumor organoid, antibody-dependent cellular cytotoxicity, 3D cell-analysis system

## Abstract

Patient-derived tumor organoids (PDOs) represent a promising preclinical cancer model that better replicates disease, compared with traditional cell culture models. We have established PDOs from various human tumors to accurately and efficiently recapitulate the tissue architecture and function. Molecular targeted therapies with remarkable efficacy are currently in use against various tumors. Thus, there is a need for in vitro functional-potency assays that can be used to test the efficacy of molecular targeted drugs and model complex interactions between immune cells and tumor cells to evaluate the potential for cancer immunotherapy. This study represents an in vitro evaluation of different classes of molecular targeted drugs, including small-molecule inhibitors, monoclonal antibodies, and an antibody-drug conjugate, using lung PDOs. We evaluated epidermal growth factor receptor and human epidermal growth factor receptor 2 (HER2) inhibitors using a suitable high-throughput assay system. Next, the antibody-dependent cellular cytotoxicity (ADCC) activity of an anti-HER2 monoclonal antibody was evaluated to visualize the interactions of immune cells with PDOs during ADCC responses. Moreover, an evaluation system was developed for the immune checkpoint inhibitors, nivolumab and pembrolizumab, using PDOs. Our results demonstrate that the in vitro assay systems using PDOs were suitable for evaluating molecular targeted drugs under conditions that better reflect pathological conditions.

## 1. Introduction

Molecular targeted therapy is one of the most important paradigm shifts in the history of cancer therapy. Traditional anticancer chemotherapeutic agents block cell division and DNA replication, and reduce the size of tumors. Although chemotherapeutic agents lead to an extension of patients’ overall survival, they are not effective for all types of cancer and induce side effects. Recently, molecular targeted drugs have been developed that interfere with specific molecules to block cancer growth, progression, and metastasis [1,2,3]. Many molecular targeted drugs have demonstrated remarkable clinical success in treating myriad types of cancer, including breast, leukemia, colorectal, lung, and ovarian cancer. In addition, targeting the immune system, which accelerates anti-tumor activity through immune checkpoint inhibition, is proving to be an increasingly effective method for treating various cancers, prolonging life, and increasing progression-free survival [1,2,3]. However, molecular targeted approaches continue to be limited by wide variations in the degree and durability of patient responses and side effects, and numerous cancers remain completely refractory to such therapy. Thus, molecular targeted therapy needs further improvement for greater clinical efficacy.

Historically, human cancer cell lines have been widely used for studies as preclinical models to evaluate anticancer agents. However, these models may not reflect the characteristics of the source tumor tissues in vivo, as they are frequently passaged for long periods of time, which may lead to alterations in their genome sequences, gene-expression profiles, and morphologies. In addition, almost all cell lines are cultured under monolayer conditions or used as xenografts in mice, which is not physically representative of tumor tissues [4,5]. Therefore, the results of evaluations performed with cancer cell lines do not accurate predict the clinical effects of anticancer drugs. Indeed, ~85% of preclinical agents entering oncology clinical trials fail to demonstrate sufficient safety or efficacy required to gain regulatory approval [6,7,8]. In vitro systems, including patient-derived tumor cell, organoid, or spheroid models that accurately recapitulate tissue architecture and function, have been developed for various types of tumor tissues (e.g., colon, lung, pancreatic, prostate, endometrial, liver, bladder, breast, brain, kidney, endometrium, and stomach), as have high-throughput assay systems for using these systems [9,10,11,12,13,14,15,16,17,18,19,20]. These models are promising in terms of facilitating a better understanding of cancer biology and for evaluating drug efficacy in vitro.

Previously, we established a novel series of patient-derived tumor organoids (PDOs) from various types of tumor tissues from the Fukushima Translational Research Project, which are designated as Fukushima (F)-PDOs. F-PDOs could be cultured for >6 months and formed cell clusters with similar morphologies to their source tumors [21]. Comparative histological and comprehensive gene-expression analyses also demonstrated that the characteristics of PDOs were similar to those of their source tumors, even following long-term expansion in culture. In addition, suitable high-throughput assay systems were constructed for each F-PDO in 96- and 384-well plate formats. We suggest that assay systems based on F-PDOs may be utilized to evaluate anticancer agents under conditions that better reflect clinical conditions (compared with conventional methods using cancer cell lines) and to discover markers of the pharmacological effects of anticancer agents.

Although several cell-based assay systems using cancer cells have been developed for evaluating molecular targeted drugs, more efficient and simple cell-based assay systems for identifying clinically efficacious therapy potency are desired. To address this issue, we have attempted to construct efficient cell-based assays for evaluating molecular targeted drugs including small molecules, monoclonal antibodies, and immune-checkpoint inhibitors using F-PDOs, which maintain the characteristics of their source tumors. In this study, epidermal growth factor receptor (EGFR) and human epidermal growth factor receptor 2 (HER2) inhibitors, including small molecules, monoclonal antibodies, and antibody-drug conjugates (ADCs) in clinical use, were evaluated using lung F-PDOs. EGFR is a tyrosine kinase receptor, and its activation triggers the activation several downstream pathways including the RAS/mitogen-activated protein kinase (MAPK), phosphoinositide 3-kinase (PI3K)/Akt, and Janus kinase (JAK)/signal transducer and activator of transcription protein (STAT) pathways that regulate cell proliferation, survival, adhesion, migration, and differentiation [22,23,24,25]. EGFR overexpression and EGFR-mediated signaling-pathway dysregulation have been observed in tumors from patients with several cancers, especially non-small cell lung cancer. Thus, several anti-EGFR drugs have been developed for clinical use in cancer therapy, including the small-molecule tyrosine kinase inhibitors (TKIs) gefitinib, erlotinib, afatinib, neratinib, dacomitinib, and osimertinib, as well as the monoclonal antibodies cetuximab and necitumumab [22,23,24,25]. In addition, HER2 can activate the same key signaling pathways as EGFR. Thus, HER2 is an ideal target for anticancer agents, and several HER2 inhibitors including trastuzumab and pertuzumab (monoclonal antibodies); afatinib, lapatinib, and neratinib (TKIs); and trastuzumab emtansine (an ADC) have been developed and approved for clinical use [24,26]. Therefore, we first evaluated the functional potency of several EGFR and HER2 inhibitors by high-throughput screening (HTS) using three lung F-PDOs (RLUN5, RLUN16, and RLUN21). Moreover, the antibody-dependent cellular cytotoxicity (ADCC) activity of trastuzumab was evaluated using three lung F-PDOs, and the complex interactions of immune cells with F-PDOs during ADCC responses to trastuzumab were visualized.

In addition, a system for evaluating immune-checkpoint inhibitors using lung F-PDOs was developed, based on non-invasive, label-free, and real-time cellular impedance monitoring technology (xCELLigence) [27] to measure the potencies of biologics. Recently, Cerignoli et al. developed the real-time, impedance-monitoring system to assess immune-checkpoint inhibitors using prostate cancer PC3 cells, in combination with an anti- programmed cell death-1 (PD-1) antibody and peripheral blood mononuclear cells (PBMCs) [27]. Therefore, we utilized the xCELLigence platform to investigate whether F-PDOs are also suitable for assessing the effects of checkpoint inhibitors on the immune system. Interactions between PD-1 expressed on the surface of activated T cells and its ligand PD-L1 on cancer cells inhibit the ability of T cells to attack the target cells. Immune-checkpoint inhibitors remove inhibitory signals of T-cell activation, which enables tumor-reactive T cells to overcome regulatory mechanisms and mount an effective antitumor response; thus, they are now used clinically for treating a broad range of tumor types [28,29]. We selected the anti-PD-1 monoclonal antibodies, nivolumab and pembrolizumab, as representative immune-checkpoint inhibitors in this study.

## 2. Materials and Methods

### 2.1. Antibodies, Compounds, and Reagents

Trastuzumab, pertuzumab, and trastuzumab emtansine were purchased from Roche (Basel, Switzerland). Cetuximab and nivolumab were obtained from Bristol-Myers Squibb (New York, NY, USA). Pembrolizumab was provided by Merck & Co. (Kenilworth, NJ, USA). An anti-Ki-67 antibody (ab16667) was purchased from Abcam (Cambridge, UK).

Seventy-eight anticancer agents were tested in this study (Appendix A). All compounds were dissolved in dimethyl sulfoxide at a concentration of 20 mM and stored at −80 °C until use. The purity and integrity of the compounds were measured via ultra-performance liquid chromatography-mass spectrometry (Waters Corporation, Milford, MA, USA), using a 1-µL injection volume, as follows. A Waters CORTECS C_18_ column (particle size: 1.6 µm; column size: 2.1 × 50 mm; Waters Corporation) was developed with a linear aqueous acetonitrile (MeCN) gradient containing a 0.1% formic acid (5–90% MeCN, 1.6 min; flow rate, 1 mL/min), separation was performed at 40 °C, and the components of the major ultraviolet (UV) adsorption peaks were verified by mass spectrometry (Appendix A).

Epidermal growth factor (EGF) and interferon-γ (IFN-γ) were obtained from Fujifilm Wako Pure Chemical, Ltd. (Osaka, Japan). Staphylococcal enterotoxin B (SEB) was obtained from Sigma-Aldrich (St. Louis, MO, USA).

### 2.2. Cells

All experiments with human material were performed in accordance with the guidelines of the Declaration of Helsinki and were approved in advance by the ethics committee of the Fukushima Medical University (approval number: 1953; approval date: 21 October 2018). Clinical specimens used for the PDOs were acquired from cancer patients at Fukushima Medical University Hospital after providing informed consent. Previously, we reported the establishment of 53 F-PDOs from human tumor tissues [21]. Out of those PDOs, three lung F-PDOs (RLUN5, RLUN16, and RLUN21) were selected for use in this study. Briefly, RLUN5, RLUN16, and RLUN21 were each established from lung cancer tissues removed from the lung by surgery. As described previously [21], the cancer tissues were washed with Hank’s balanced salt solution (Fujifilm Wako Pure Chemical, Ltd.) supplemented with penicillin-streptomycin solution (Fujifilm Wako Pure Chemical, Ltd.), cut with a scalpel into small pieces (approximately 1 mm^3^), and cultured in suspension for 3–6 months to establish the F-PDOs.

THP-1 cells were obtained from Japanese Collection of Research Bioresources Cell Bank (Osaka, Japan). PBMCs were provided by Precision Bioservices (Frederick, MD, USA).

### 2.3. Cell Culture

F-PDOs were cultured at 37 °C in 15 mL FBIM001 [21] using ultra-low attachment 75-cm^2^ flasks (Corning, Inc., NY, USA) in a humidified incubator with 5% CO_2_. Because accurate cell numbers of F-PDOs could not be determined using a cell counter, the volumes of cell pellets after centrifuging cell suspensions in 15-mL tubes were visually estimated by comparison with tubes marked at volumes of 75, 100, and 150 µL. The 80% medium was changed twice weekly. When F-PDOs reached their maximum saturation density, the cells were passaged at a 1:2 ratio. The procedure for handling F-PDOs was described in detail elsewhere [21]. F-PDOs were cultured in flasks until sufficient cell-pellet volumes were obtained for the assay. For all experiments, the F-PDO culture media were replaced at 1 day prior to seeding, as previously described [21].

THP-1 cells were cultured in vendor-recommended growth media supplemented with fetal bovine serum (FSA; Sigma-Aldrich, St. Louis, MO, USA) and penicillin-streptomycin solution (Fujifilm Wako Pure Chemical, Ltd.) at final concentrations of 10 and 1%, respectively, at 37 °C in a humidified incubator with 5% CO_2_. Cell numbers and viabilities were automatically measured using trypan blue dye exclusion with a Vi-Cell XR Cell Viability Analyzer (Beckman Coulter, Inc., Brea, CA, USA), according to the manufacturer’s protocol.

### 2.4. Real-Time PCR

Relative levels of mRNA were quantified with the StepOnePlus Real-Time Polymerase Chain Reaction System (Thermo Fisher Scientific, Waltham, MA, USA) using TaqMan Fast Virus 1-Step Master Mix (Thermo Fisher Scientific) and TaqMan Gene Expression Assays (assay #Hs01076090_m1 for EGFR, assay #Hs01001580_m1 for HER2, and assay #Hs99999901_s1 for the 18S ribosomal gene as the internal control; Thermo Fisher Scientific) according to the manufacturer’s protocols.

### 2.5. Three-Dimensional Cell Analysis

Images were captured using the FLUOVIEW FV3000 Confocal Laser Scanning Microscope (Olympus, Tokyo, Japan). All imaging data were analyzed with the NoviSight 3D Cell Analysis System (Olympus). Immunofluorescence staining of F-PDOs was performed using an anti-Ki-67 antibody (1:250 dilution) and 4′,6-diamidino-2-phenylindole (DAPI) after fixation in phosphate buffer solution containing 4% paraformaldehyde (PFA; Fujifilm Wako Pure Chemical, Ltd.) and 1% Triton X-100 (Fujifilm Wako Pure Chemical, Ltd.). The F-PDOs were then incubated for 5 h at room temperature in AbScale solution, comprised of Dulbecco’s phosphate-buffered saline (D-PBS (–); Fujifilm Wako Pure Chemical, Ltd.) containing 0.33 M urea and 0.1–0.5% Triton X-100 [30], to wash the cells. Next, the F-PDOs were incubated for 30 min at room temperature in AbScale rinse solution, comprised of 0.1× D-PBS (–) containing 2.5% BSA and 0.05% (*w/v*) Tween-20 [30]. The rinse solution was exchanged with 4% PFA, and the F-PDOs were incubated at room temperature for 30 min.

Cetuximab and trastuzumab were labeled using the HiLyte Fluor 555 Labeling Kit-NH_2_ (Dojindo Laboratories, Kumamoto, Japan) and used to quantify the cellular expression levels of EGFR and HER2. In detail, F-PDOs were incubated for 3 h with 10 µg/mL fluorophore-labeled antibodies in 24-well, flat-bottomed, ultra-low-attachment microplates (Corning, Inc.). Cells were then washed with D-PBS (–) and fixed overnight at 4 °C in 4% PFA. Subsequently, the F-PDOs were washed twice with D-PBS (–) and imbedded with 7.5% acrylamide gel. The F-PDOs were washed with D-PBS (–) and stained overnight at 4 °C with DAPI. Finally, the F-PDOs were incubated overnight at room temperature in SCALEVIEW-S4 (Fujifilm Wako Pure Chemical, Ltd.).

To determine whether the effector cells could access the F-PDOs via trastuzumab, F-PDOs were treated with antibodies and then co-cultured with THP-1 effector cells that were pre-stimulated with IFN-γ. Twenty-four hours before treating F-PDOs with antibodies, IFN-γ (10 ng/mL) was added to the THP-1 cells, and the F-PDOs were seeded in 24-well plates, as described above. On the following day, the F-PDOs were treated with 10 or 100 µg/mL of HiLyte Fluor 555-labeled antibodies for 30 min. Next, the THP-1 cells were rinsed with medium to wash away the IFN-γ and stained using the Cell Explore Fixable Live Cell Tracking Kit Green Fluorescence (AAT Bioquest, Inc., Sunnyvale, CA, USA). The fluorescently labelled THP-1 cells were added to the F-PDO target cells at a density of 1 × 10^5^ cells per well and co-cultured for 3 or 120 h. The cells were fixed overnight with 4% PFA at 4 °C, washed three times with D-PBS (–), imbedded with 7.5% acrylamide gel, and stained with DAPI in D-PBS (–) containing 0.1% Triton X-100. Finally, the F-PDOs were incubated overnight at room temperature in SCALEVIEW-S4.

### 2.6. Cell-Viability Assay

The cell viability of F-PDOs was assayed using a previously reported method [21]. Briefly, each F-PDO was minced using a CellPet FT (JTEC Corporation, Osaka, Japan) equipped with a filter holder containing a 70- or 100-µm mesh filter. Each F-PDO suspension was diluted 20-fold and seeded into 384-well, round-bottomed, ultra-low-attachment microplates (Corning, Inc.) in 40 μL medium, using a Multidrop Combi Reagent Dispenser (Thermo Fisher Scientific, Inc.). At 24 h after seeding, the F-PDOs were treated with 0.04-μL solutions of different compounds using an Echo 555 Liquid Handler (Labcyte, Inc., San Jose, CA, USA) or 0.8-μL solutions of different antibodies using an ADS-348-8 Multistage-Dispense Station (Biotec Co., Ltd., Tokyo, Japan). The cells were treated with compounds or antibodies at final concentrations ranging from 1.0 nM to 20 µM or 0.195 to 100 µg/mL, respectively, using a series of 10 concentrations (serially diluted 3-fold) in each case. After 144 h, 10 μL CellTiter-Glo 3D solution (Promega Corporation, Madison, WI, USA) was added to F-PDOs in each well, and the plates were mixed using a mixer and incubated for 15 min at 25°C. Luminescence was measured using an EnSpire Plate Reader (PerkinElmer, Inc., Waltham, MA, USA). Cell viability was calculated by dividing the amount of ATP in the test wells by that in the vehicle-control wells, after subtracting the background levels. The growth rate over 6 days was calculated by dividing the amount of ATP in the wells without anticancer agents by those in the vehicle-control wells 24 h after seeding.

The half-maximal inhibitory concentration (IC_50_) and area under the activity curve measuring dose response (AUC) values (used to measure dose–response relationships) were calculated from the dose–response curves and analyzed using Morphit software, version 6.0 (The Edge Software Consultancy, Ltd., Guildford, UK). As the first approach, the response curves were fitted to the luminescence signal intensities using a 4-parameter sigmoid model. Alternatively, a sigmoidal fixed-slope model without a Hill equation was used. The data shown represent the mean ± standard deviation of triplicate experiments. The Z′ factor, a dimensionless parameter that ranges between 1 (infinite separation) and < 0, was defined as Z′ = 1 – (3σc^+^ + 3σc^–^)/|µc^+^ – µc^–^|, where σc^+^, σc^−^, µc^+^, and µc^−^ are the standard deviations (σ) and averages (µ) of the high (c^+^) and low (c^–^) controls [31].

### 2.7. Lactate Dehydrogenase (LDH) Assay

One day before cell seeding, 96-well microplates (Sumitomo Bakelite Co., Ltd., Tokyo, Japan) were coated with 50 µL fibronectin (50 µg/mL), and THP-1 cells were stimulated by adding 10 ng/mL IFN-γ. Each F-PDO was minced using the CellPet FT. Each F-PDO suspension was diluted 10-fold, and a 100-μL volume of each suspension was seeded into the pre-coated plates. At 24 h after seeding the F-PDOs, 60 µL of growth medium was removed and 10 µL of antibody solution was added at a final concentration of 10 or 100 µg/mL. THP-1 effector cells were added to each F-PDO at 2 × 10^4^ cells/well 30 min after adding the antibody. Each well contained a final volume of 100 µL. After 3 or 120 h, 25 μL of supernatant was recovered, mixed with 25 μL of reagent solution from the Cytotoxicity Detection Kit^PLUS^ (Roche Diagnostics, Rotkreuz, Switzerland), and incubated for 15 min. Stop solution (12.5 μL) was added to the reaction mixture and mixed using a mixer for 10 s. Absorbance was measured at 490 and 690 nm using an EnSpire Plate Reader. Cytotoxicity was calculated by dividing the LDH activity in the test wells by that in the vehicle-control wells, after subtracting the background absorbance levels, expressed as a percentage.

### 2.8. Real-Time Potency Assessment Using the xCELLigence RCTA System

The xCELLigence RTCA System (ACEA Bioscience, San Diego, CA, USA) was used for evaluating the immune-checkpoint inhibitors. The E-plate 96 (ACEA Bioscience), which is a single-use 96-well plate specialized for performing cell-based assays with the xCELLigence RTCA System, was coated overnight with fibronectin (0.5 µg/well) at 4 °C. After removing the fibronectin, 50 µL of culturing medium was added to each well of the E-plate to measure the background impedance. Then, each F-PDO was minced using the CellPet FT. The F-PDO suspension was diluted 15-fold and 50 µL of each cell suspension was seeded into the wells. The plate was placed in a safety cabinet at room temperature for 30 min and transferred to the xCELLigence RTCA instrument at 37 °C in a CO_2_ incubator. At 24 h after seeding the F-PDOs into the E-plate, 60 µL of culture media was removed from each well, and 10 µL of antibody solution was added. PBMC were stimulated with 5 ng/mL SEB for 24 h and then added to F-PDOs 30 min post-antibody treatment. PBMCs were added at 1 × 10^4^ cells per well to each F-PDO. Each well contained a final volume of 100 µL. The impedance was measured every 15 min. Changes in impedance signals were measured as the cell index and then converted to percent-cytolysis values using xCELLigence immunotherapy software (ACEA Bioscience). “Percent cytolysis” refers to the percentage of target cells that were killed by effector cells, checkpoint inhibitors, or both when compared to RLUN16 cells alone (as a control). The cell indexes of wells containing PBMCs alone was subtracted from the cell indexes of the sample wells, for each time point. Next, each value was normalized to the cell index at the time just before antibody addition. The normalized cell index was converted to % cytolysis using xCELLigence immunotherapy software according to the following equation: % cytolysis = (1 – normalized cell index [sample wells])/normalized cell index (target alone wells) × 100.

## 3. Results and Discussion

### 3.1. Lung F-PDOs

The RLUN5, RLUN16, and RLUN21 lines, established from lung cancer tissues, were used in this study. RLUN5 was derived from lung cancer tissue, which was pathologically diagnosed as adenosquamous carcinoma mixed with squamous cell carcinoma and solid adenocarcinoma. RLUN16 and RLUN21 were derived from squamous cell carcinoma specimens.

RLUN5 was observed to be a mixture of cells with different morphological features (Figure 1a). The majority of RLUN5 cells appeared as cell clusters that were ~100–300 µm in diameter. RLUN16 was comprised primarily of round single cells after passaging that formed big sheets ranging from 200 to >1000 µm in diameter (Figure 1a, lower panel). The sheets were easily disrupted by pipetting or shaking (Figure 1a, upper panel). RLUN21 grew as large, firm, high-density cell clusters of 300–1000 µm in diameter that frequently merged to form clusters that were >1000 µm in diameter (Figure 1a). The doubling times of RLUN5, RLUN 16, and RLUN 21 were 5, 7, and 10 days, respectively.

The cell number, cell volume, cell density, and Ki67-positive cell number in four representative cell clusters of each F-PDO were analyzed with NoviSight (Figure 1b–e). The total cell number in each cell cluster was calculated by counting the number of nuclei stained with DAPI. The median volume of RLUN21 cell clusters was larger (12 × 10^6^ voxels) than those of RLUN5 (2.8 × 10^6^ voxels) and RLUN 16 (0.8 × 10^6^ voxels) (Figure 1b,c). RLUN16 clusters were very small because the big sheets readily dissociated during the experiments. The cell densities of RLUN5 and RLUN16 were almost identical (approximately 3 × 10^−4^/voxel), whereas that for RLUN21 was approximately 2 × 10^−4^/voxel (Figure 1d). Thus, the cell density of RLUN21 was lower than that of RLUN5 and RLUN16. Next, proliferating cells were identified by detecting Ki67, a cellular marker of proliferation. The numbers and percentages of Ki67-positive cells in cell clusters were analyzed (Figure 1b,e). Ki67-postive cells were located on the surfaces of the cell clusters and, indicating that cell proliferation occurred primarily on the surface of each cell cluster. The percentages of Ki67-positive cells varied among the three F-PDOs. The ratio of Ki67-positive cells was highest for RLUN16, at approximately 40%. The ratio for RLUN21, which proliferated very slowly, was very low (2%). The characteristic structure of RLUN21 is shown in a three-dimensional (3D) video (Video S1). By performing 3D cell analysis with NoviSight, it was possible to accurately investigate the structural features of F-PDOs possessing various structures, unlike conventional 2D analysis.

To evaluate the pharmacological activity of drugs targeting EGFR and HER2 with lung F-PDOs, the gene-expression levels of EGFR and HER2 in RLUN5, RLUN16, and RLUN21 were examined. The mRNA expression level of EGFR in RLUN21 was approximately 400-fold higher than those of RLUN5 and RLUN16 (Figure 2a). However, no difference was observed in the HER2 gene-expression levels among the three lung F-PDOs (Figure 2b).

The protein-expression levels of EGFR and HER2 in the three lung F-PDOs were detected by immunofluorescence using the monoclonal anti-EGFR antibody cetuximab and the monoclonal anti-HER2 antibody trastuzumab. The mean red-fluorescence intensity (corresponding to EGFR expression) in RLUN21 was much higher than those in RLUN5 and RLUN16, consistent with mRNA-expression analysis, and the entire surfaces of the cell clusters were strongly stained (Figure 2c). Approximately half of the cells in RLUN21 clusters were EGFR-positive (Figure 2e). Next, we confirmed the expression of the HER2 protein in these F-PDOs. The percentage of HER2-positive cells (red fluorescence) in RLUN21 clusters (65%) was >6 times that in the RLUN5 (11%) and RLUN16 (3%) clusters. The results were not identical to those for HER2 mRNA expression. HER2 was also expressed heterologously on the surface of the F-PDO clusters. Heterologous expression of HER2 is a feature of F-PDO, which is an important insight gained from 3D cell analysis.

We examined whether EGF, a ligand for EGFR, affected RLUN21 proliferation because RLUN21 showed high EGFR expression. The F-PDOs were cultured in medium with or without EGF for six days. The cell-growth rate was calculated by measuring the ATP contents to count viable cells. As expected, RLUN21 proliferation increased 2.6-fold in medium with EGF and, thus, was dependent on EGF (Figure 3a,b). However, RLUN5 and RLUN16 proliferation did not change after EGF was deed to the culture medium. Next, the number of Ki67-positive cells in RLUN21 cell clusters was analyzed by immunofluorescence (Figure 3c). The positive-cell ratios in RLUN21 cell clusters cultured with EGF were two-fold higher than those in culture without EGF (Figure 3d). In addition, the cell clusters cultured with EGF showed a lower cell density and were loose (Figure 3e). These results suggest EGF induced RLUN21 cell division by activating the EGFR signal pathway.

Taken together, these data indicate that F-PDOs displayed various distinct structural and biological characteristics when compared to conventional cell lines cultured in monolayers.

### 3.2. Evaluation of Small-Molecule Drugs Using F-PDOs

To determine the sensitivity of the lung F-PDOs to anticancer agents including (EGFR and HER2 inhibitors) using our HTS system with 384-well plates, growth inhibition was assessed using RLUN5, RLUN 16, and RLUN 21. EGFR inhibitors are used as standard clinical treatments for lung cancer. F-PDOs were treated with EGFR inhibitors 24 h post-seeding and were subsequently incubated for six days. The ATP contents were measured to count the viable cells. The HTS performance was evaluated by computing the Z′ factor. The Z′ factor has been widely accepted as a means for evaluating assay quality and performance, and an assay is considered suitable for HTS when the Z′ factor value is >0.5 [31]. The F-PDOs showed 2.7–5.8 fold growth in the plates. The control data points in the 384-well plate assay had calculated Z′ factor values of 0.63, 0.94, and 0.62 for RLUN5, RLUN16, and RLUN21, respectively. These results suggested that this assay performed well for HTS, using 384-well plates to evaluate anticancer agents.

The half-maximal inhibitory concentration (IC_50_) and area under the activity curve measuring dose response (AUC) values of anticancer agents observed with each F-PDO are presented in Figure 4 and Appendix A. AUC values do not require extrapolation and can be estimated from dose-response curves. Therefore, AUC values are often used to evaluate the efficacies of anticancer agents. The AUC values demonstrated that three F-PDOs were similarly sensitive to anti-cancer drugs. RLUN21, which is dependent on EGFR signaling pathway, was more sensitive to inhibitors targeting members of the EGFR signaling pathway than other F-PDOs (Figure 4a). In particular, RLUN21 proliferation was slowed more by the inhibitors when cultured with EGF than without EGF (Figure 4a). These results confirmed that RLUN21 proliferation was highly dependent on the EGFR pathway.

RLUN5 showed higher sensitivity to chemotherapeutic agents than other F-PDOs, whereas RLUN21 showed resistance to chemotherapeutic agents (Figure 4c). In particular, the microtubule-targeting cytotoxic agents, paclitaxel, and vindesine, did not significantly inhibit the proliferation of RLUN21, which proliferated very slowly and did not possess Ki67-positive cells (Figure 1b,e). However, RLUN21 clusters were more sensitive to microtubule-targeting cytotoxic agents in the presence of EGF than in the absence of EGF. These results imply that antimitotic agents strongly act on actively dividing cells.

The F-PDO donors had no history of EGFR inhibitor administration. Therefore, the effects of EGFR inhibitors against PDOs could not be compared to assess the clinical efficacies. However, the donor of RLUN16 was previously administered cisplatin and vinorelbine (a microtubule-targeting agent) after surgery, and had no recurrence of cancer. RLUN16 showed high sensitivity to paclitaxel and vindesine. Thus, the results suggest that the evaluation of anticancer drug using F-PDO reflects the clinical efficacy. Overall, these results demonstrated the suitability of the HTS system for evaluating various anticancer agents including molecular targeted and chemotherapeutic drugs against lung F-PDOs in 384-well plate format.

### 3.3. Evaluation of Antibody Drugs Using F-PDOs

#### 3.3.1. Evaluation of Monoclonal Antibodies and an ADC

Growth inhibition was assessed using the HTS system with 384-well plates to investigate the sensitivity of three F-PDOs to monoclonal antibody drugs for cancer. To this end, we tested cetuximab (which targets EGFR); trastuzumab (which targets HER2); pertuzumab (which inhibit the dimerization of HER2 with other HER homologs); and trastuzumab emtansine (which is an ADC consisting of trastuzumab covalently linked to the microtubule-targeting cytotoxic agent DM1 that targets HER2). F-PDOs were treated with the indicated drugs 24 h post-seeding and were subsequently incubated for six days. The viable F-PDO cells were counted. Cetuximab inhibited the proliferation of RLUN21 in a dose-dependent manner with an IC_50_ value of 2.4 µg/mL, whereas the IC_50_ values were >100 µg/mL for RLUN5 and RLUN16, which are not dependent on EGF for proliferation (Table 1). These results suggest that inhibiting EGFR signaling in RLUN21 with cetuximab induced growth inhibition due to high EGFR expression and EGF dependency for proliferation in RLUN21.

Trastuzumab did not inhibit the growth of RLUN5, RLUN16, or RLUN21, indicating that trastuzumab could not neutralize these F-PDOs. However, the IC_50_ values of an ADC, trastuzumab emtansine, were 3.4, 4.9, and 1.7 µg/mL for RLUN5, RLUN16, and RLUN21, respectively, and the ADC showed strong cytotoxic effects against each F-PDO. The values were weekly correlated with the protein-expression level of HER2 (Figure 2f). Even RLUN16 with weak HER2 expression (2.8% of the all cells were HER2-positive) showed a high effect of the ADC due to a bystander effect, wherein a cytotoxic agent released from cancer cells penetrated the cell membranes of neighboring dividing cancer cells and exerted further cytotoxic effects [32]. In addition, trastuzumab emtansine, an ADC that functions as a microtubule-targeting cytotoxic agent, strongly induced cell death in RLUN21 despite the resistance of RLUN21 to microtubule-targeting cytotoxic agents (Figure 4c). These results suggest that ADCs may serve as effective cancer drugs in clinical use.

#### 3.3.2. ADCC Activity of Trastuzumab Against RLUN21

Trastuzumab did not show neutralizing activity against the F-PDOs used in this study (Table 1). To confirm this lack of ADCC activity against F-PDOs, we tested effector cell-mediated cytolysis with the anti-HER2 antibody trastuzumab in RLUN21 cells, which overexpressed HER2. ADCC is a defense mechanism involving cellular immunity, whereby effector cells of the immune system actively lyse cancer cells, whose membrane-surface antigens are bound by specific antibodies. To measure ADCC activities, the monocyte cell line THP-1 was used as the effector cells, and cytotoxicity was measured as the activity of extracellularly released LDH from F-PDOs when killed by THP-1 cells. The percentage of cytolysis was detected by measuring the LDH activity, and THP-1 cells were used as effector cells. RLUN21 cells treated with 100 µg/mL trastuzumab showed significantly more cytolysis in the presence of THP-1 effector cells than in the absence of THP-1 cells at 120 h after THP-1 cell addition, although no such effect was observed with 10 µg/mL trastuzumab or at 3 h after THP-1 addition (Figure 5a).

Next, we observed interactions between F-PDO and THP-1 effector cells via trastuzumab. At 3 h after co-culturing RLUN21 and THP-1 cells, THP-1 cells had begun to gather around the RLUN21 cell clusters via trastuzumab (Figure 5b). THP-1 cells invaded RLUN21 cell clusters treated with 100 µg/mL trastuzumab at 120 h, but RLUN21 clusters treated with 10 µg/mL trastuzumab did not show differences after 3 h. These results imply that the invasion of THP-1 cells into the cell clusters causes the cytolysis of RLUN21.

Taken together, these results revealed that these F-PDOs can be used for evaluating the efficacies of neutralizing activity, ADCC activity, and ADC.

### 3.4. Evaluation of Immune-Checkpoint Inhibitors Using F-PDOs

We performed experiments using RLUN16 and PBMCs respectively as target and effector cells, as well as nivolumab and pembrolizumab, which are monoclonal antibodies targeting PD-1 as immune checkpoint inhibitors. We selected the xCELLigence platform, which can be used to monitor the number, morphology, and attachment of cells for a long duration. To induce the adequate expression of PD-1 in PBMCs for the immune-checkpoint experiments, the effector cells were treated with the bacterial superantigen SEB (5 ng/mL) beginning at one day before adding the target cells and were continuously stimulated thought the measurements. RLUN16 cells were treated with SEB-stimulated PBMCs in the presence of nivolumab and pembrolizumab. As shown in Figure 6, SEB-stimulated PBMCs reduced RLUN16 cell proliferation by only 20%, compared with control cells. The percent cytolysis of RLUN16 cells following treatment with nivolumab (Figure 6a) or pembrolizumab (Figure 6b) alone was almost identical to that after co-culture with PBMCs. In contrast, the presence of nivolumab and pembrolizumab led to a significant increase in PBMC-mediated RLUN16 cell killing. The cytolytic activity in the presence of 1 µg/mL pembrolizumab was greater than that observed in the presence of 50 µg/mL nivolumab. This result may reflect differences in the binding affinities of pembrolizumab and nivolumab to PD-1 [33]. The K_d_ value (i.e., the equilibrium dissociation constant between the antibody and its antigen) of pembrolizumab is 0.028 nM, while nivolumab is 2.6 nM [33]. Thus, a 100-fold difference in the K_d_ values was consistent with the biological evaluations using F-PDOs. In summary, these results demonstrate that the F-PDO assay system with real-time, impedance-based technology can be utilized for assessing immune-checkpoint inhibitors.

## 4. Conclusions

In conclusion, we developed systems for assaying molecular target drugs, antibody drugs, ACDs, and immune-checkpoint inhibitors using F-PDOs, which are characteristically similar to the source tissues. In addition, our results indicated that F-PDOs have structural characteristics enabling the construction of an evaluation system for anticancer drugs, based on structural changes of F-PDO found with a 3D cell-analysis system, demonstrating that 3D cell analysis is a powerful tool for analyzing the characteristics of PDOs. Finally, the results of this study demonstrate that F-PDOs with various distinct structural and biological characteristics are superior for identifying potential novel molecular target drugs for cancer.

## Figures and Tables

**Figure 1 cells-08-00481-f001:**
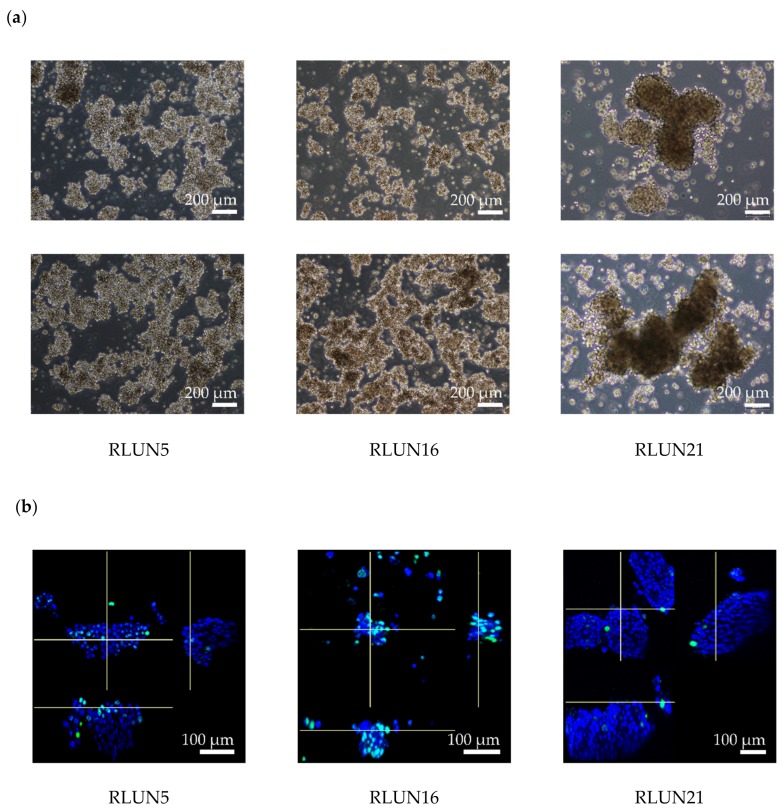
Phase-contrast and confocal images of three lung F-PDOs. (**a**) Phase-contrast images of RLUN5, RLUN16, and RLUN21 were obtained using a ×4 objective. Scale bar: 200 µm. The upper panels show each F-PDO immediately after passage and the lower panels show each F-PDO before passage. (**b**) Confocal images of RLUN5, RLUN16, and RLUN21 were obtained using a ×10 objective. Scale bar: 100 µm. Ki67-expression images for RLUN5, RLUN16, and RLUN21, prepared using an anti-Ki67 antibody (green). DNA stained with DAPI (blue). (**c**–**e**) Confocal imaging data analyzed with NoviSight. (**c**) Box-and-whisker plot of the cell cluster volumes. (**d**) Box-and-whisker plot of the cell density expressed as the cell number/cell volume in a cell cluster. (**e**) Box-and-whisker plot of the Ki67-positive ratio expressed as 100× the number of positive cells/the number of total cells in a cluster.

**Figure 2 cells-08-00481-f002:**
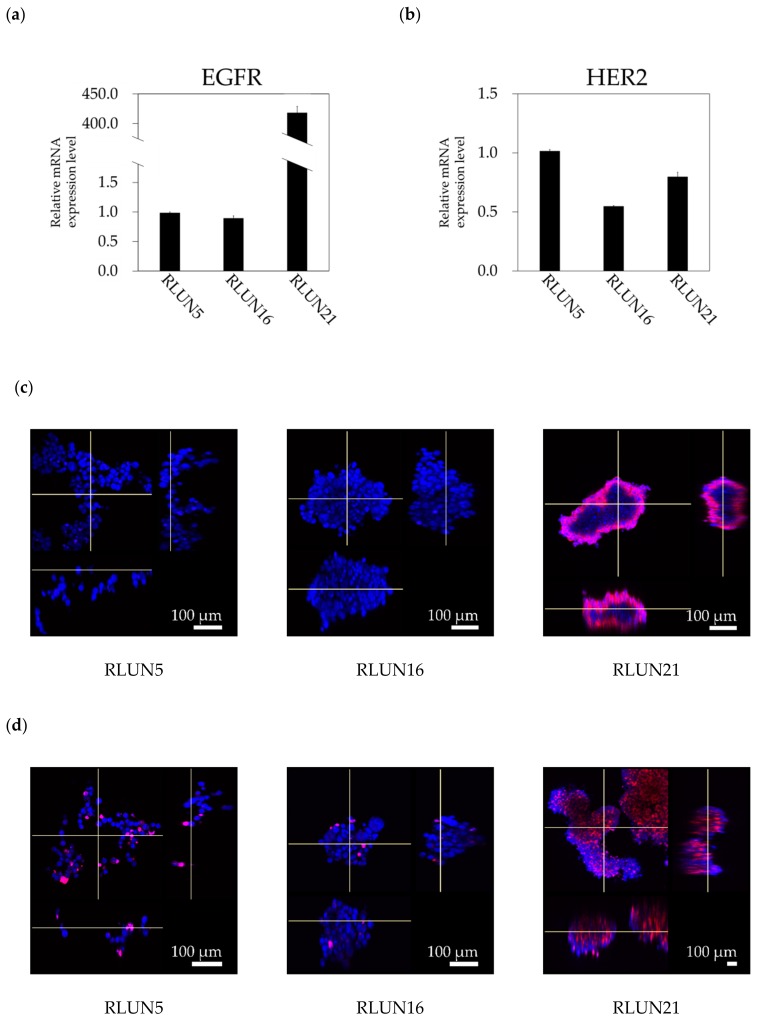
Expression of epidermal growth factor receptor (EGFR) and human epidermal growth factor receptor 2 (HER2) in three lung F-PDOs. mRNA-expression levels of (**a**) EGFR and (**b**) HER2 in RLUN5, RLUN16, and RLUN21. Gene-expression levels were normalized against that of 18S rRNA gene, using the 2^−ΔΔCt^ method. The expression data for RUNL16 and RUNL21 were normalized to those for RLUN5, which was set to 1. The error bars indicate the standard deviation from three replicate samples. (**c**) EGFR-expression images of RLUN5, RLUN16, and RLUN21 stained with fluorescently labeled cetuximab (red). (**d**) HER2-expression images stained with fluorescently labeled trastuzumab (red). DNA stained with DAPI (blue). Magnification: ×10. Scale bar: 100 µm. Box-and-whisker plots of the ratio of cells positive for EGFR (**e**) or HER2 (**f**), calculated as 100× volume of positive cells/volume of cells in a cluster.

**Figure 3 cells-08-00481-f003:**
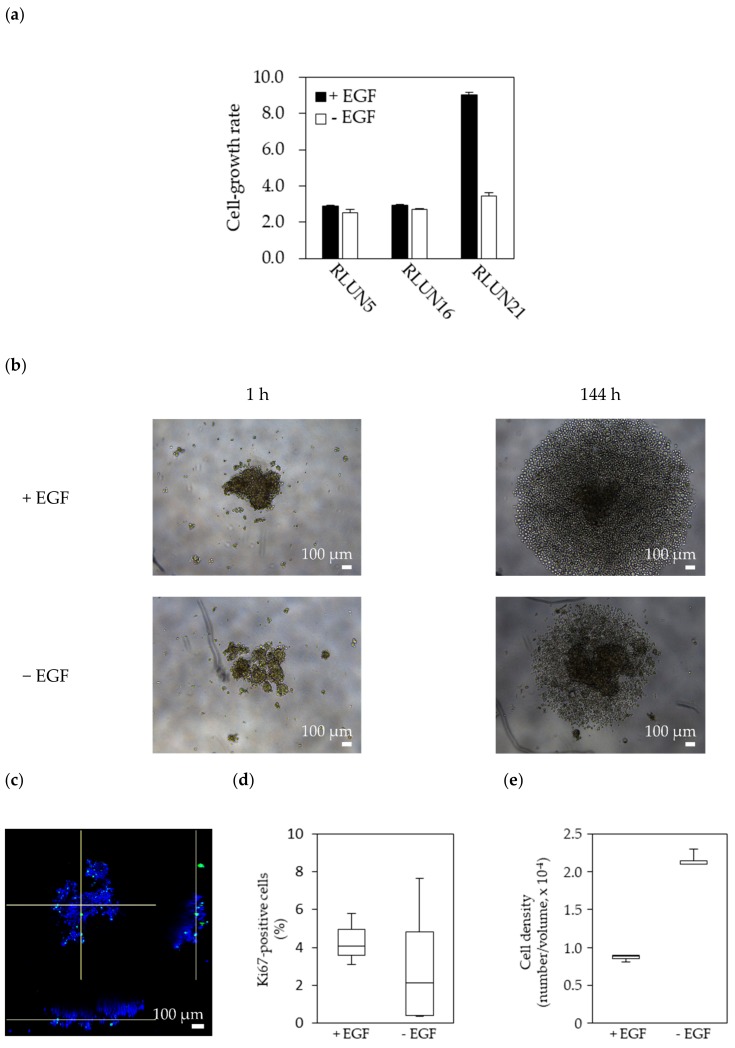
Dependency of F-PDO proliferation on epidermal growth factor (EGF). (**a**) Growth rate of RLUN5, RLU16, and RLUN21 in the presence or absence of EGF. Each F-PDO was seeded in a 96-well, round-bottomed, ultra-low-attachment microplate (Corning, Inc.). Twenty-four hours after seeding, EGF was added into the wells at a final concentration of 100 ng/mL. The wells without EGF were also set up to confirm the dependency of cell growth on EGF. One hour (culture start time) or 144 h later, the amount of ATP was measured as described in Section 2.6. Growth rates were calculated by dividing the amount of ATP in each well at 144 h by that at the culture start time. (**b**) Phase-contrast images of RLUN21 cultured with EGF (upper panels) or without EGF (lower panels) for six days. The images were obtained using a ×5 objective. Scale bar: 100 µm. (**c**) Images of Ki67 expression in RLUN21 cultured with EGF were obtained using an anti-Ki67 antibody (green). DNA stained with DAPI (blue). Magnification: ×10. Scale bar: 100 µm. Box-and-whisker plot of the Ki67-positive ratio (**d**) and cell density (**e**) of RLUN21 cells cultured with EGF.

**Figure 4 cells-08-00481-f004:**
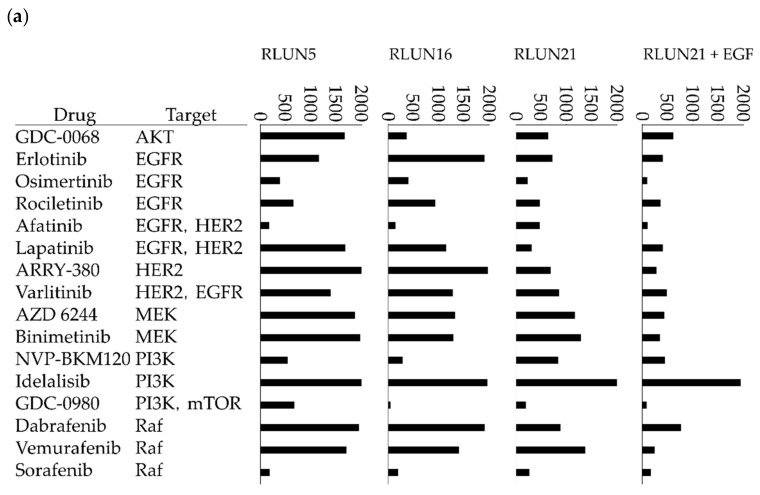
Response of F-PDOs to anti-cancer agents. The bar graphs represent the area under the activity curve measuring dose response (AUC) values calculated from the growth-inhibition assays, using 78 anticancer agents at concentrations ranging from 3 nM to 20 µM. The AUC values show the area under each fitting curve calculated from each dose-response curve. The AUC value of a sample not treated with an anti-cancer drug was 2000. Low AUC values indicate cases where F-PDOs were more sensitive to an anti-cancer drug. Values above 2000 indicate cases where the cells grew more than the untreated sample. The data shown represent the mean of triplicate experiments. The column entitled “RLUN21 + EGF” indicates cases where the F-PDOs were grown in the presence of EGF. (**a**) Molecular target drugs that inhibit the EGFR signal pathway. (**b**) Other molecular target drugs. (**c**) Chemotherapeutic drugs.

**Figure 5 cells-08-00481-f005:**
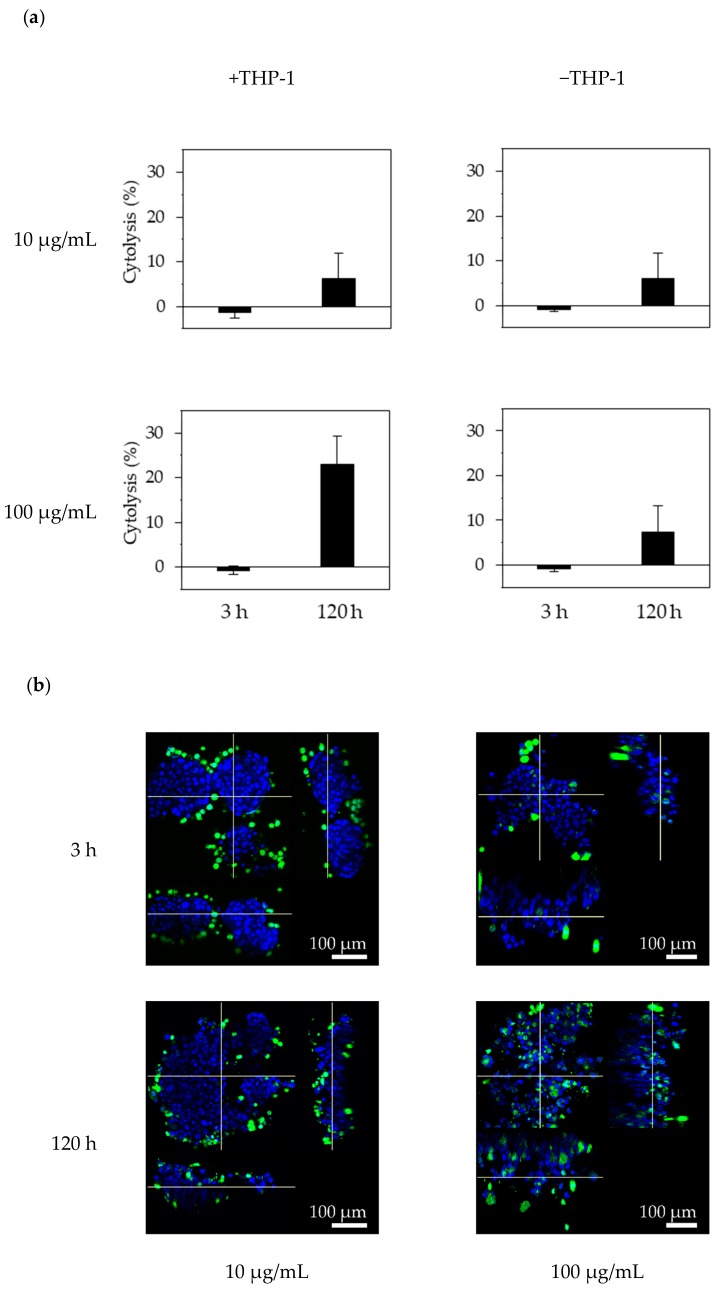
Antibody-dependent cellular cytotoxicity (ADCC) activity against RLUN21 cells with trastuzumab. (**a**) Cytolysis against RLUN21 by trastuzumab and THP-1 effector cells was determined by performing lactate Dehydrogenase (LDH) assays. Cytolysis (%) was calculated as 100 × (LDH release in test well –LDH release from RLUN21 cells only –LDH release from THP-1 cells only)/(maximum LDH release from RLUN21 cells lysed using Triton X-100 –LDH release from RLUN21 cells only). Cytolysis was measured at 3 and 120 h after THP-1 addition. The error bars indicate the standard deviation from six replicate samples. (**b**) Interaction of THP-1 effector cells with RLUN21 cell during ADCC. Magnification: ×30. Scale bar: 100 µm. THP-1 cells stained using the Cell Explore Fixable Live Cell Tracking Kit are shown in green. DNA stained with DAPI is shown in blue.

**Figure 6 cells-08-00481-f006:**
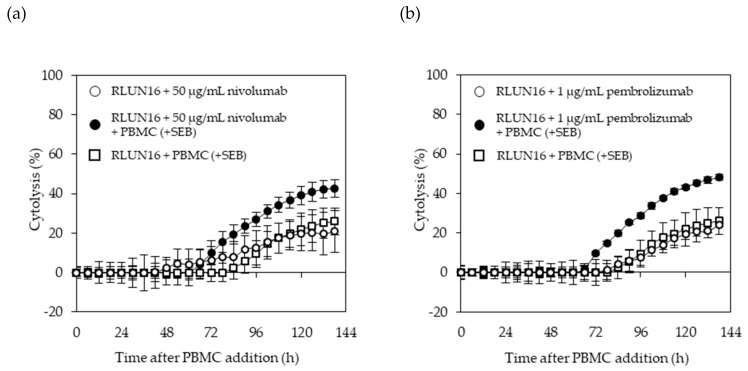
Impedance measurements of RLUN16 cell cytolysis by nivolumab (**a**) or pembrolizumab (**b**). Open circles, anti-PD-1 antibody alone; closed circles, anti-PD-1 antibody in combination with Staphylococcal enterotoxin B (SEB)-stimulated peripheral blood mononuclear cells (PBMCs); open squares, SEB-stimulated PBMCs alone. Nivolumab and pembrolizumab were used at concentrations of 50 and 1 μg/mL, respectively. Dynamic changes in the cell-index values were recorded over time.

**Table 1 cells-08-00481-t001:** Half-maximal inhibitory concentration (IC_50_) values of trastuzumab, pertuzumab, cetuximab, and trastuzumab emtansine against each F-PDO.

	RLUN5	RLUN16	RLUN21
Cetuximab	>100	>100	2.4
Trastuzumab	>100	>100	>100
Pertuzumab	>100	>100	>100
Trastuzumab emtansine	3.4	4.9	1.7

IC_50_ values reflect concentrations in term of µg/mL.

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
