# Peer review of "An In Vitro System for Evaluating Molecular Targeted Drugs Using Lung Patient-Derived Tumor Organoids"

_cells, 2019, doi:10.3390/cells8050481_

Round 1
Reviewer 1 Report
Dear Authors,
I found the article very interesting and well written, and will recommend for publication.
I would suggest some improvements:
The figure 4b and c may benefit from sorting compounds by potency or other criteria. The figure legend may include more technical details, which would be easier for reader to follow. Also, the data from individual experiments should be shown in supplementary materials.
The methods and results section may benefit from better description of the details of proliferation assay and THP-1 assay.
Author Response
1. The figure 4b and c may benefit from sorting compounds by potency or other criteria. The figure legend may include more technical details, which would be easier for reader to follow. Also, the data from individual experiments should be shown in supplementary materials.
Ans: According to reviewer’s suggestion, we revised Figure 4b and c, the figure legend and the results (3.2. Evaluation of small-molecule drugs using F-PDOs). Also, we added the IC50 and AUC values in supplementary materials.
2. The methods and results section may benefit from better description of the details of proliferation assay and THP-1 assay.
Ans: We indicated the details of proliferation assay in the legend of Figure 3. According to reviewer’s suggestion, we added the details of THP-1 assay in the results (3.3.2. ADCC activity of trastuzumab against RLUN21).
Reviewer 2 Report
In this study, the authors investigated EGFR and HER2 inhibitors, nivolumab, and pembrolizumab using an assay system, and the antibody-dependent cellular cytotoxicity (ADCC) activity was visualized to show the interactions between immune cells and PDOs during ADCC responses. Although the results of various researches positively support the author's claim, there are some concerns that must be considered.
1. In the methodology, clinical procedures for acquiring individual tissues from lung cancer patients are missing.
2. The authors must provide etiology information for individual patients. General patients with lung cancer are treated with EGFR-targeted drugs such as gefitinib or erlotinib as a first line therapies for standard regimens. Since the EGFR mutations may be present after target therapy, the adequacy of this study can be determined by the treatment information.
3. Herceptin is known as a therapeutic agent for breast cancer patients. This reviewer wonder why the authors investigated the efficacy of traditional lung cancer target therapies with those of breast cancer treatment.
4. The cell cytotoxicity by increase in the lactate dehydrogenase response is judged from the results of supernatant analysis in a cell culture environment. The increase in cytoplasmic lactate dehydrogenase is also associated with common cancer cell metabolism.
5. Regretfully, this reviewer constantly doubt whether the drug screening system established by the authors can be applicable to a variety of genetic mutations.
Author Response
1. In the methodology, clinical procedures for acquiring individual tissues from lung cancer patients are missing.
Ans: According to reviewer’s suggestion, we added the clinical procedures in the 2.2 cells, Materials and Methods.
2. The authors must provide etiology information for individual patients. General patients with lung cancer are treated with EGFR-targeted drugs such as gefitinib or erlotinib as a first line therapies for standard regimens. Since the EGFR mutations may be present after target therapy, the adequacy of this study can be determined by the treatment information.
Ans: According to reviewer’s suggestion, we added the etiology information and discussion of drug treatment in the results (3.2. Evaluation of small-molecule drugs using F-PDOs).
3. Herceptin is known as a therapeutic agent for breast cancer patients. This reviewer wonder why the authors investigated the efficacy of traditional lung cancer target therapies with those of breast cancer treatment.
Ans: Trastuzumab is used clinically for HER2 overexpressing breast and stomach cancer. Although lung cancer is not clinically applicable, the effect of trastuzumab was examined because HER2 was overexpressed in the lung F-PDOs.
4. The cell cytotoxicity by increase in the lactate dehydrogenase response is judged from the results of supernatant analysis in a cell culture environment. The increase in cytoplasmic lactate dehydrogenase is also associated with common cancer cell metabolism.
Ans: It is common to examine LDH activity in culture media to detect cell cytotoxicity by ADCC activity. We added the details of ADCC assay in the results (3.3.2. ADCC activity of trastuzumab against RLUN21).
5. Regretfully, this reviewer doubts whether the drug screening system established by the authors can be applicable to a variety of genetic mutations.
Ans: We are currently performing whole exome analysis of all F-PDOs and analyzing gene mutation information. Therefore, it will be possible to evaluate the efficacy of anticancer drugs based on gene mutations in the future.